# The Influence of a High-Cholesterol Diet and Forced Training on Lipid Metabolism and Intestinal Microbiota in Male Wistar Rats

**DOI:** 10.3390/ijms25105383

**Published:** 2024-05-15

**Authors:** Yuliya S. Sidorova, Nikita A. Petrov, Yuliya M. Markova, Alexey I. Kolobanov, Sergey N. Zorin

**Affiliations:** Federal Research Centre of Nutrition and Biotechnology, 109240 Moscow, Russia; petrov-nikita-y@mail.ru (N.A.P.); yulia.markova.ion@gmail.com (Y.M.M.); alleexxkl@yandex.ru (A.I.K.); zorin@ion.ru (S.N.Z.)

**Keywords:** cholesterol, hypercholesterolemia, physical activity, lipid metabolism, liver, intestinal microbiota, rats

## Abstract

Adequate experimental animal models play an important role in an objective assessment of the effectiveness of medicines and functional foods enriched with biologically active substances. The aim of our study was a comparative assessment of the effect of consumption of 1 or 2% cholesterol with and without regular (two times a week), moderate running exercise on the main biomarkers of lipid and cholesterol metabolism, as well as the intestinal microbiota of male Wistar rats. In experimental rats, a response of 39 indicators (body weight, food consumption, serum biomarkers, liver composition, and changes in intestinal microbiota) was revealed. Total serum cholesterol level increased 1.8 times in animals consuming cholesterol with a simultaneous increase in low-density lipoprotein cholesterol (2 times) and decrease in high-density lipoprotein cholesterol (1.3 times) levels compared to the control animals. These animals had 1.3 times increased liver weight, almost 5 times increased triglycerides level, and more than 6 times increased cholesterol content. There was a tendency towards a decrease in triglycerides levels against the background of running exercise. The consumption of cholesterol led to a predominance of the *Bacteroides* family, due to a decrease in *F. prausnitzii* (1.2 times) and bifidobacteria (1.3 times), as well as an increase in *Escherichia* family (1.2 times). The running exercise did not lead to the complete normalization of microbiota.

## 1. Introduction

Obesity is a heterogeneous condition characterized by excessive accumulation of fat in various fat depots. Under conditions of excess energy (nutrients), its excess accumulates in white adipocytes of subcutaneous adipose tissue (SAT), becoming both a reserve that can be used up in conditions of energy deficiency and protection against cooling. Meanwhile, in the conditions of modern life, these functions of SAT have ceased to be relevant for most people, and since the consumption of fat reserves from the depot practically does not occur due to the absence of energy deficiency, its reserves become excessive, exceeding the depositing capacity of cells [1,2]. According to the World Health Organization, overweight and obesity are the fifth leading cause of death in the world. The leading cause of death in humans from obesity is associated with an increased risk of a number of concomitant chronic diseases, such as type 2 diabetes, insulin resistance, cardiovascular disease, non-alcoholic fatty liver disease, and especially hyperlipidemia [3,4].

People with hyperlipidemia have more than twice the increased risk of developing cardiovascular disease (including myocardial infarction and stroke) than people with normal cholesterol levels [5]. Hyperlipidemia is an imbalance in the blood levels of cholesterol, including low-density lipoprotein cholesterol (LDL) and high-density lipoprotein cholesterol (HDL). Other forms of hyperlipidemia include hypertriglyceridemia, as well as mixed hyperlipidemia, in which both cholesterol and triglyceride levels are elevated [5,6]. Under the excess intake of cholesterol with food, its production by the liver is reduced, and the excess is excreted by the intestines. If this process is disrupted, the level of cholesterol in the blood increases pathologically, which leads to hypercholesterolemia development. The implementation of cholesterol homeostasis largely depends on the amount and spectrum of steroids, as well as other lipids included in food, the intensity of endogenous cholesterol synthesis, its absorption from the digestive tract, destruction and transformation into other compounds by tissue and microbial enzymes, relationships with bile acids, the intensity of their enterohepatic circulation, the amount of excretion in feces, hormonal status, and other factors. High-fat diets, especially rich in saturated fatty acids and refined carbohydrates, fried meats, high-cholesterol foods, and sugary drinks significantly increase cholesterol levels [7]. Clinical and experimental studies showed that a high-fat (HFD) and/or high-cholesterol diet can cause non-alcoholic fatty liver disease in both obese and non-obese people [8,9].

The intestinal microbiota is known to be involved in the regulation of lipid metabolism and plays a critical role in maintaining homeostasis of the intestine–liver axis [10,11,12]. Food lipids can influence the intestinal microbiota through fatty acids formed from lipid precursors under the action of lipases. These fatty acids may have antibacterial activity or can be used by intestinal bacteria as metabolic substrates [12]. On the other hand, the intestinal microbiota produces different metabolites that influence lipid metabolism in the host organism (primary and secondary bile acids, short-chain fatty acids, trimethylamine N-oxide) [13,14,15]. The abnormal levels of blood lipids are usually associated with severe dysbiotic disorders of the intestinal microbiota, with a decrease in lacto- and bifidobacteria quantity [16] and reduced microbial diversity [14]. The microbiome is considered to be the connecting factor between metabolic disorders, obesity, insulin resistance (IR), dyslipidemia, type 2 diabetes mellitus, hypertension, and cardiovascular disease. Non-alcoholic fatty liver disease (NAFLD) is the most common chronic liver disease, despite the well-defined mechanisms of its pathogenesis, there is increasing evidence supporting the hypothesis of the intestinal microbiota’s role in the development of this disease. Intestinal dysbiosis and increased permeability of the intestinal epithelium lead to translocation of microbial components, in particular lipopolysaccharides of Gram-negative bacteria (endotoxins) and β-glucan of fungi, which are collectively termed microbial or pathogen-associated molecular patterns (MAMPs/PAMPs). These patterns are recognized by immune receptors on liver cells, which initiate and maintain inflammatory cascades that eventually lead to liver damage in the form of fibrosis [10,17]. In addition, the mechanisms by which the intestinal microbiota may influence the progression of NAFLD include increased energy production, modulation of choline and bile acid metabolism, and increased production of endogenous ethanol [18].

Some of the ways to prevent obesity and its complications are changing eating habits, adequate nutrition, using functional foods enriched with biologically active substances, and increasing physical activity. The results of preclinical in vivo studies on experimental animal models play an important role in an objective assessment of the effectiveness of functional products enriched with biologically active food substances from the standpoint of evidence-based medicine. The quality of scientific research is largely determined by the methodology that the researcher uses. The more difficult the studied problem is, the higher the importance of the methodological correctness of the work performed.

The following protocols are used to model in vivo hyperlipidemia: the addition of 1 or 2% cholesterol, as well as 1–2% cholesterol + 0.5% cholic acid to the animal diet without or with a change in the fat component. At the same time, there is not much information about the effect of the hypercholesterolemic diet itself on the animals since this diet is usually used as a positive control [19]. Thus, the aim of our study was a comprehensive comparative assessment of the effect of consumption of 1% and 2% cholesterol with and without regular physical activity on mature male Wistar rats, and the search for valid biomarkers characterizing disturbances in the functional state of the animal’s body, lipid and cholesterol metabolism, liver condition, as well as the intestinal microbiota of the animals.

## 2. Results

We conducted preliminary testing of animals in an open field to divide them according to individual behavioral differences. This approach can increase the degree of verifiability and reliability of the results. The general condition of all animals in appearance, fur quality, food and water consumption, and behavior during daily inspection was satisfactory.

### 2.1. Body Weight and Food Intake

Figure 1 shows the average cumulative feed consumption of animals for the entire experiment and the dynamics of body weight gain of animals throughout the experiment.

Animals of group 1% Chol consumed significantly more food compared to animals of the Control group and animals of group 2% Chol. Animals of groups 1% Chol + RUN and 2% Chol + RUN, exposed to running exercise, consumed significantly more food compared to groups Control and 2% Chol.

From the 14th day of the experiment, the body weight gain of 2% Chol + RUN animals was significantly lower compared to group 1% Chol + RUN. The absolute body weight of all animals did not differ significantly throughout the experiment and was 465 ± 10; 470 ± 19; 464 ± 18; 484 ± 13; and 458 ± 13 g on day 94, respectively. The increase in body weight at the end of the experiment also did not differ significantly between the groups.

### 2.2. Body Composition

Table 1 presents the results of studying the body composition of experimental animals.

The body fat mass of animals in groups 1% Chol, 1% Chol + RUN, and 2% Chol + RUN) significantly decreased on the 33rd day of the experiment compared to the beginning of the experiment, remained almost unchanged until the end of the experiment and was significantly lower both in comparison with control animals and in comparison with animals of group 2% Chol.

On the 82nd day of the experiment, the body composition of animals in group 2% Chol changed significantly compared to intact animals in group Control: the percentage of fat in these animals became significantly higher, while the percentage of lean body mass and free water decreased significantly.

### 2.3. Elevated Plus Maze

Figure 2 shows the results of the elevated plus maze test.

Animals of groups 1% Chol + RUN and 2% Chol + RUN, exposed to daily physical activity, spent significantly more time in the open arms of the maze and significantly less time in the closed arms of the maze compared to animals in the Control group. The obtained result indicates the influence of running loads against the background of a high-cholesterol diet on the anxiety component, characterized by the time spent in the open and closed arms of the maze, without changing motor activity (Figure 2C,D).

### 2.4. Insulin Resistance Test

We detected no significant differences in the glucose level of all animals (5.5 ± 0.2; 5.1 ± 0.3; 5.4 ± 0.2; 5.2 ± 0.2; and 5.3 ± 0.2 mmol/L, respectively), on the 75th day of the experiment.

Figure 3 shows the results of the insulin resistance test.

The obtained values of area under the curve (AUC) indicators (Figure 3B) indicate the absence of the development of insulin resistance in rats of all experimental groups on the background of consumption of exogenous insulin compared with rats of the Control group. At the same time, in animals of groups 1% Chol, 1% Chol + RUN, and 2% Chol + RUN, the AUC indicator was slightly lower (at the level of a trend) than in animals of groups Control and 2% Chol.

Figure 3A reflects the changes in the blood glucose level of animals after insulin administration over time. The glucose level of 2% Chol animals decreased less than in other animals, which may indicate the beginning of the development of insulin resistance in these animals.

### 2.5. Blood and Liver Biochemical Analysis

Table 2 and Table 3 and Figure 4 present the results of determining the general biochemical analysis of animal blood serum, the levels of SOD, MDA, leptin, and ghrelin in the blood serum of animals, as well as indicators of lipid metabolism in the liver of animals.

A significant increase in the levels of low-density lipoproteins (LDL) was noted in the blood of animals of group 1% Chol along with a decrease in the levels of high-density lipoproteins (HDL) compared to the Control group. Also, the level of total bilirubin was significantly reduced in these animals.

Significantly higher levels of globulins and albumin were noted in the blood of animals of group 2% Chol compared to the Control group. Consumption of 2% cholesterol in the diet led to a significant increase in the blood levels of cholesterol and LDL of animals compared to animals of the Control group without changes in HDL levels. Animals consuming 2% cholesterol showed a significant increase in the levels of total bilirubin, AST, and ALT compared to the Control group.

The increase in the LDL level against the background of the decrease in HDL level was noted in the blood of animals of group 1% Chol + RUN compared to animals of the Control group. The level of total bilirubin in these animals was significantly reduced, and the level of AST was significantly increased compared to the Control group.

Significantly higher levels of globulins and albumin were noted in the blood of animals of group 2% Chol + RUN, similar to group 2% Chol. Cholesterol and LDL levels were also increased compared to Control. A significantly lower level of bilirubin was revealed in comparison with animals of the Control group. Additionally, an increase in creatinine and alkaline phosphatase was detected in these animals.

Our study did not reveal a deficiency of detectable minerals in the blood serum of animals. Only a significant increase in the level of calcium in the blood of animals of 2% Chol and 2% Chol + RUN groups compared to animals in the control group was shown.

All animals that received exogenous cholesterol with food showed a significant increase in both absolute and relative liver weight. The significantly higher accumulation of fat (more than two times) and cholesterol (more than five times) in the liver of these animals was shown compared with animals of the Control group. The accumulation of triglycerides was not significant only for group 1% Chol + RUN, which received a low dose of cholesterol against the background of physical exercise. There was a tendency towards a decrease in liver triglyceride levels against the background of physical exercise.

The animals of groups 1% Chol and 2% Chol receiving cholesterol had a significant increase in SOD content with a significant increase in the accumulation of MDA in the blood serum compared to animals of the Control group. We observed compensation for these parameters in animals of groups 1% Chol + RUN and 2% Chol + RUN, which received cholesterol against the background of running exercise: neither the SOD level nor the MDA level differed significantly from the control.

We did not detect any significant changes in the serum ghrelin level. At the same time, a significant decrease in leptin levels was shown in animals of 1% Chol and 1% Chol + RUN groups. On the contrary, we see an increase in serum leptin in the animals of group 2% Chol, which may indicate the development of resistance to this hormone. The animals of group 2% Chol + RUN showed a trend towards a decrease in blood leptin, which indicates that physical exercise can prevent the development of leptin resistance.

### 2.6. Blood Hematological Analysis

Table 4 presents the results of a general whole-blood analysis of experimental animals.

The significant decrease in the absolute values of leukocytes, lymphocytes, monocytes, and granulocytes in the blood, as well as the average volume of erythrocytes, can be noted in animals of the experimental group 2% Chol compared to animals of group 1% Chol. The animals in this group showed a decrease in the average erythrocyte volume, average hemoglobin content in the erythrocyte, and hematocrit compared to animals in the Control group.

Only a significant decrease in hematocrit was detected in animals of groups 1% Chol and 1% Chol + RUN, compared to the Control group.

### 2.7. Study of the Microbiota

Intestinal microbiota population levels are presented in Table 5. The results obtained indicate the effect of both doses of cholesterol on the intestinal microbiota.

A significant increase in the content of *Escherichia coli* was detected in animals of groups 1% Chol, 2% Chol, and 1% Chol + RUN compared to Control animals, while the changes for group 2% Chol + RUN were characterized as a trend (*p* < 0.1).

There was a decrease in the levels of *Faecalibacterium prausnitzii* against the background of cholesterol consumption, and this was statistically significant for animals of groups 1% Chol, 2% Chol and 2% Chol + RUN, however for animals of group 1% Chol + RUN a trend towards similar changes was revealed.

Also, animals of groups 1% Chol, 2% Chol, and 2% Chol + RUN had signs of anaerobic imbalance, namely, an increase in the ratio of *Bacteroides* spp. and *F. prausnitzii*. At the same time, the indicated ratio in group 1% Chol + RUN was significantly different from the values in group 1% Chol but did not differ from the values in the Control group.

In addition, cholesterol consumption led to an increase in the content of lactobacilli: in animals of the 1% Chol group there was a tendency to increase (*p <* 0.1), and for the remaining groups of animals this increase was significant.

In animals of group 1% Chol, an increase in the content of *Blautia* spp., and *Ruminococcus* spp., belonging to the *Clostridia* class, was detected and in the other experimental groups, there was a tendency (*p <* 0.1) to an increase in the content of *Blautia* spp. and a slight increase in the frequency of occurrence of *Ruminococcus* spp.

There was a decrease in the level of bifidobacteria in animals of groups 1% Chol + RUN and 2% Chol + RUN compared to the Control, and animals in group 2% Chol showed a tendency (*p <* 0.01) to a decrease in this protective population.

## 3. Discussion

As a part of broad efforts to combat non-communicable diseases including nutrition-dependent ones, the main targets are to identify the principal risk factors and to find ways to prevent, treat, and manage the disease. The results of in vivo preclinical studies using experimental animal models undoubtedly play an important role. The aim of our study was to find the simplest, valid, and, most importantly, reproducible model that can adequately reflect the course of the disease to further ensure its correction.

When assessing the validity and the possibility of further successful use of any model, one should take into account that the extreme effects applied on the animal lead to the development of reversible changes but do not cause the development of severe pathological disorders. To ensure successful correction of the modeled disease, including nutrition and physical activity factors, which are the most accessible and economically acceptable forms of behavior, we chose a model with unchanged proportions of the main energy sources in the diet. The administration of exogenous cholesterol in two dosages, most often found in the literature (1 and 2%) was considered a factor of malnutrition. Additionally, we assessed the influence of moderate physical activity on the formation of changes.

We obtained no changes in body weight and body weight gain in animals, which is generally consistent with the literature data [20]. In [21], adult Sprague Dawley rats (n = 8) were fed a normal diet containing 1% cholesterol and 0.5% cholic acid for 4 weeks, and there were no changes in body weight, growth, or food intake. Male Sprague Dawley rats (n = 15) received a standard diet supplemented with 1% cholesterol and 0.3% sodium cholate for 16 weeks; no significant differences in body weight were shown [22]. A similar study found no weight gain over 16 weeks [23]. However, in our study, using the impendansometry method, we found that when consuming 2% cholesterol, there was a significant accumulation of fat in the rat’s body, which was accompanied by a loss of lean mass. Thus, although we do not see a difference in weight and growth between animals, the process of obesity development can be considered as developing.

There is also a modification of the hypercholesterolemia model where cholesterol is introduced into a high-fat or high-carbohydrate diet. The study [24] on growing male Wistar rats also showed that consumption of a high-fat diet (13% fat) with the addition of 1% cholesterol led to an increase in body weight in rats (n = 5) after 30 days of the experiment. A high-fat diet (60 kcal fat), supplemented with 1.25% cholesterol and 0.5% cholic acid, caused an increase in body weight gain in female SD rats (n = 6) only at 8 weeks of feeding [20].

Our own studies [25] also showed an increase in body weight in male Wistar rats fed a high-fat high-carbohydrate diet supplemented with 2% cholesterol for 90 days. This modeling method is limited by the severe and possibly irreversible disorders developing in the animal’s body due to the strong exogenous cholesterol load, which is expressed primarily by the extreme accumulation of fat, cholesterol, and triglycerides in the liver. There are concerns that further successful correction of these disorders will be impossible, especially using nutritional approaches.

Changing only the fat or carbohydrate component in the animal’s diet does not always lead to excessive weight gain, requires a long period of exposure, and is difficult to reproduce, which corresponds to the high adaptive ability of animals of this species [26,27].

The main biomarkers of hyperlipidemia development in vivo are total cholesterol, LDL and HDL cholesterol, and triglycerides [28].

In our study, animals consuming a normal diet supplemented with 1% cholesterol had a statistically significant decrease in serum HDL and an increase in serum LDL compared to intact animals. The levels of total cholesterol and triglycerides did not change. Supplementation with 2% of cholesterol led to an increase in total cholesterol along with an increase in LDL. The running exercise did not have any beneficial effect on the lipid profile. The work [22] also revealed only a significant increase in total cholesterol along with LDL growth, without changes in serum triglyceride levels in animals receiving 1% cholesterol for 16 weeks. In the study [23], a significant increase in total cholesterol was detected only by the 16th week of feeding. There were no changes in blood triglycerides.

Total blood bilirubin is a promising biomarker, from the point of view of its influence on the processes of inhibition of LDL oxidation [29]. It is known that oxidative modification of human low-density lipoproteins (LDL) is involved in the formation of plaques in blood vessels, thereby significantly increasing the risks of developing cardiovascular diseases and type 2 diabetes [30,31]. The introduction of cholesterol into the diet of animals, regardless of the dose and the presence/absence of training, led to a significant decrease in the level of total bilirubin.

Non-alcoholic fatty liver disease is the most common chronic liver disease and is often caused by excessive consumption of fatty foods and foods containing high concentrations of cholesterol and is mainly treated through lifestyle changes [32]. NAFLD progresses sequentially: it begins with the deposition of fat in the liver (steatosis), then excessive accumulation of fat in the liver leads to an inflammatory process (steatohepatitis), then fibrosis occurs, and the final stage develops cirrhosis [33]. It is the early stage of NAFLD that is reversible. That is why it is most rational to start treatment at the stage of steatosis, before the development of steatohepatitis and fibrosis. Regular exercise or physical activity can help reduce fat deposition in the liver by increasing energy expenditure, improving skeletal lipid oxidation, and reducing body weight [34].

Non-alcoholic fatty liver disease usually develops in patients with overweight, obesity, type II diabetes, high cholesterol, or high blood triglyceride levels. Serum markers may show slight increases in the levels of transaminases (ALT predominates), alkaline phosphatase, and gamma-glutamyl transpeptidase; bilirubin levels rarely increase [35]. Moreover, changes in these indicators are not specific and do not correlate with the severity of the disease. No elevation of AST and ALT enzymes can be observed in 80% of patients with NAFLD.

Animal studies on the pathophysiology of NAFLD have been widely conducted in recent years, mainly in dietary modification models (high fructose, high fat, cholesterol and cholic acid supplemented diets, methionine/choline-deficient diets) [36,37], drug models (streptozotocin, carbon tetrachloride, diethylnitrosamine) and genetic models of laboratory animals.

The protocol we used for the administration of 1 and 2% cholesterol showed a significant increase in indicators characterizing changes in the condition of the liver: an increase in liver weight, accumulation of fat, triglycerides, and cholesterol in the liver, against the background of consumption of both cholesterol doses, and no dose-dependent effect was revealed. Only the introduction of 2% cholesterol led to a significant increase in both indicators of AST and ALT, while running training reduced the ALT content to the level of the normal control.

Running load against the background of a small dose of cholesterol led to a decrease in the liver triglycerides level to the level of the control group; when consuming a high dose of cholesterol, a tendency towards a decrease in this indicator can also be noted. Authors of the study [38] showed that intense exercise had a greater effect on liver vacuolation density and lipid reduction in male Wistar rats fed a high-fat diet than moderate exercise. Serum and liver triglyceride and total cholesterol levels also demonstrated different sensitivity to exercise intensity.

Oxidative stress and lipid peroxidation are involved in the pathogenesis of many diseases, including atherosclerosis, coronary heart disease, diabetic angiopathy, and NAFLD [39]. An increased content of free fatty acids may cause liver dysfunction, since fatty acids are chemically active and stimulate free radical oxidation of biomolecules. The resulting reactive oxygen species (ROS) can damage the biological membranes of liver cells. Reactive oxygen species attack polyunsaturated fatty acids and initiate lipid peroxidation within the cell, leading to the formation of aldehyde byproducts such as MDA [40]. Normally, this process is physiologically balanced due to the activity of endogenous antioxidant systems, which are able to increase activity in response to an increase in pro-oxidant effects.

Consumption of 1% and 2% cholesterol led to a statistically significant accumulation of MDA and SOD in the blood serum of animals. The observed increase in SOD concentration may reflect the mobilization of the antioxidant defense component in response to the development of oxidative stress in the body, reflected by the accumulation of MDA. SOD is an inducible enzyme, i.e., its synthesis increases if peroxidation is activated in cells. Accordingly, when exogenous cholesterol is consumed in both doses, there is no depletion of the antioxidant defense system. Running exercise led to a decrease in both indicators to the level of control animals.

In our study, we obtained a significant increase in animals’ appetite receiving 1% cholesterol, as well as both doses of cholesterol against the background of running exercise. According to the results of a bioimpedance analysis, a decrease in fat mass in the body composition was registered in animals of these groups. On the contrary, only animals receiving 2% cholesterol had an increase in appetite and, accordingly, an increase in the accumulation of body fat (significant compared to the control by the 82nd day of the experiment). The results obtained are consistent with changes in the serum leptin level: in animals receiving 1% cholesterol without exercise, a significant decrease in leptin was shown; in animals receiving 2% cholesterol against the background of a running exercise, no significant differences were shown with the control, although its concentration was lower at the trend level. The highest leptin levels were recorded in animals receiving 2% cholesterol.

The hormone leptin, produced by adipocytes, controls body weight, food intake, and energy expenditure. Its secretion is proportional to the amount of adipose tissue present in the body, which gives an idea of the current level of fat in the body [41]. Leptin levels increase with increasing fat mass, thereby suppressing food intake, whereas loss of fat mass leads to a decrease in leptin levels and a subsequent increase in food intake. This mechanism maintains homeostatic control of adipose tissue mass within a relatively narrow range, thereby serving an important evolutionary function [42]. Accordingly, serum leptin, as a marker of obesity development, increases (at a trend level) only with the introduction of a larger dose of cholesterol into the diet. A similar result was shown when determining insulin resistance: an increase, also at a trend level, was noted only in animals receiving 2% cholesterol. Running exercise levels out both the increase in insulin resistance and the increase in leptin level to the level of control animals.

A picture of the dysbiotic changes against the background of cholesterol consumption was revealed, consisting of a decrease in the levels of *F. prausnitzii* and an increase in the ratio of *Bacteroides* spp./*F. prausnitzii* due to increased levels of *Escherichia coli*. *F. prausnitzii* is a significant protective population of microbiota that produces short-chain fatty acids, in particular butyrate, and exhibits anti-inflammatory properties, and when their levels decrease, inflammatory processes develop [43,44]. At the same time, the predominance of bacteria of the genus *Bacteroides* over *F. prausnitzii* is considered a dysbiotic disorder of the intestinal microbiota of a pro-inflammatory nature [45]. Despite the fact that, in general, the levels of Gram-negative *E. coli* in the experimental groups did not reach too high values, an increase in the content of this population against the background of a decrease in protective populations (*F. prausnitzii*, *Bifidobacterium*) compared to the control group is a negative factor. The outer wall of Gram-negative microorganisms contains lipopolysaccharides (LPS). When the barrier function of the intestinal epithelium is impaired, an increase in LPS levels in the blood plasma can occur (so-called metabolic endotoxemia), which, in turn, leads to nonspecific systemic inflammation [46]. Physical activity only against the background of a low dose of cholesterol prevented the development of anaerobic imbalance (the ratio of *Bacteroides* spp. and *F. prausnitzii*) and prevented a pronounced decrease in *F. prausnitzii*. However, the levels of this population still did not reach the values of the control group. The increase in lactobacilli levels compared to controls in cholesterol-fed rats may be due to the fact that many members of the *Lactobacillaceae* family participate in the enterohepatic circulation and are characterized by high bile salt hydrolase activity [47,48,49]. It is known that taking probiotic lactobacilli leads to a decrease in serum LDL levels in rats [24,50,51] and in humans [52,53,54,55,56]. Consumption of a high dose of cholesterol (2%) along with or without running exercise caused changes in the microbiota similar to the lower dose of cholesterol (1%). The absence of significant differences between these groups probably indicates the absence of a dose-dependent effect. With an excess of cholesterol in the diet, dysbiotic changes develop in the intestinal microbiota of rats, characteristic of the so-called “Western” type of microbiota with a predominance of *Bacteroides*, a decrease in *F. prausnitzii* and bifidobacteria, as well as an increase in *Escherichia*. The additional energy consumption in the form of physical activity does not lead to complete normalization of microbiota.

## 4. Materials and Methods

### 4.1. Experimental Animals

The experiment was carried out for 94 days on 50 growing male Wistar rats with an initial body weight of 250 ± 5 g (age 7 weeks). The animals were obtained from the laboratory animal nursery of the Stolbovaya branch of the Federal State Budgetary Institution of Science “Scientific Center for Biomedical Technologies of the Federal Medical and Biological Agency”. Animal studies were carried out in accordance with the requirements set out in the National Standards of the Russian Federation GOST 33647-2015 “Principles of Good Laboratory Practice” [57] and GOST 33216-2014 “Guide to the care and maintenance of laboratory animals. Rules for keeping and caring for laboratory rodents and rabbits” [58] and approved by the Local Ethics Committee of the Federal State Budgetary Institution “Federal Research Center for Nutrition and Biotechnology” (Protocol No. 11 of 15 December 2021). Animals were kept under controlled environmental conditions (temperature 20–24 °C, relative humidity 30–60%, 12 h light cycle).

### 4.2. Experimental Design

The experimental design is presented in Figure 5.

#### 4.2.1. Preliminary Division of the Animals

To preliminary divide the animals into groups, the open field test was used. Testing was carried out before the start of the experiment after the animals were acclimatized for seven days. In the open field test, an animal is placed in an unfamiliar open space from which it cannot escape. An unfamiliar environment triggers a complex set of behavioral reactions that reflect anxiety and the desire to explore new territory, i.e., the animal’s behavior in the OF is determined by the ratio of defensive and exploratory tendencies. Testing was carried out under standard lighting conditions for 3 min (180 s). During testing, the following behavioral indicators were recorded: the number of zone transitions, time spent in each zone, and distance traveled. The movement of animals across the field was recorded using the Smart 3.0.04 software (Panlab, Barcelona, Spain).

The animals were randomly divided into 5 groups according to body weight and OF test results: Control (n = 10), 1% Chol (n = 10), 2% Chol (n = 10), 1% Chol + RUN (n = 10), and 2% Chol + RUN (n = 10) (Table 6).

#### 4.2.2. Animal Treatment

Animals of all groups received a standard semi-synthetic diet [59] and drinking water ad libitum during the whole experiment. Food consumption was monitored three times a week and animals were weighed once a week.

Cholesterol was added in the amount of 1 g/100 g to the diet of animals in groups 1% Chol and 1% Chol + RUN. Cholesterol in the amount of 2 g/100 g was added to the diet of animals in groups 2% Chol and 2% Chol + RUN.

Rats of experimental groups 1% Chol + RUN and 2% Chol + RUN were subjected to regular (2 days a week) running activity on a treadmill (Panlab, Spain). The running machine was equipped with 5 running belts (which allows 5 animals to be tested simultaneously) with an adjustable speed (0–150 cm/s) and slope (from −25° to 25°). The speed of the belt was gradually increased from 20 to 30 cm/s according to the scheme presented in Table 7. The running time was 10 min. The slope of the track was 0 degrees. The shock strength in the shock zone of the track was set to 0.4 mA.

Animals are forced to run by electrical shock using an electrode placed at the lower end of the track (the shock strength can be set from 0 to 2 mA). Recorded parameters are distance traveled, number of electric shocks received, and total time of shocks.

#### 4.2.3. Animal Body Composition

The body composition of rats was studied before the start of the experiment (day 0) and on days 33, 66, and 82 by magnetic resonance relaxometry using an EchoMRI-1100 analyzer (EchoMRI LLC, Houston, TX, USA). This device is a quantitative magnetic resonance system that allows you to measure in less than 1 min the total fat and lean body mass, the amount of free water, and the total volume of water in the animal’s body without anesthesia or euthanasia.

#### 4.2.4. Elevated Plus Maze

To assess the degree of anxiety and exploratory activity of the animals, on the 80th day of the experiment, an elevated plus maze test was performed. The movement of rats through the maze was recorded using the Smart 3.0.04 video system (Panlab Harvard Apparatus, Barcelona, Spain). Testing of animals was carried out during periods of their minimum daily activity (from 10.00 to 15.00). The time spent in the open and closed arms of the maze, the number of zone transitions, and the distance traveled were recorded.

On the 75th day of the experiment, fasting glucose levels were measured in blood taken from the tail vein using an electrochemical glucometer (LifeScan Inc., Milpitas, CA, USA).

#### 4.2.5. Insulin Resistance Test

When conducting an insulin resistance test on the 85th day of the experiment, animals of all experimental groups were administered insulin intraperitoneally at a dose of 0.25 U/kg. Blood glucose levels were measured before the administration of insulin solution (0 point) and after 30, 60, 120, and 180 min. Curves of the dependence of glucose levels on time after insulin administration were plotted, and the value of the area under the curve (AUC mmol/L × 180 min) was determined.

#### 4.2.6. Sample Collection

At 24 h before end of the experiment, the animals were placed in exchange cages to collect 24 h urine.

On day 94, rats of all groups (deprived by starvation for 12 h) were decapitated under light anesthesia and a postmortem examination was performed. The cecum was isolated to study the microbiota. Cecal samples were stored at −70 °C, 1 mL of blood was collected into tubes with K_2_EDTA for hematological analysis, and the rest of the blood collected after decapitation of the animal was incubated at a temperature of 2–8 °C for 3 h, centrifuged for 30 min at 3000 rpm at 4 °C, and the resulting serum was stored at −20 °C.

#### 4.2.7. ELISA Kits

At the end of the experiment, the content of ghrelin, leptin, superoxide dismutase, and MDA was determined in the blood serum of rats using competitive ELISA according to the manufacturer’s instructions (Elabscience, Houston, TX, USA).

#### 4.2.8. Serum and Liver Biochemical Analysis

The content of indicators of protein metabolism (total protein, albumin, globulins, urea, creatinine), lipid metabolism (total cholesterol, HDL cholesterol, LDL cholesterol, triglycerides), carbohydrate metabolism (glucose), purine metabolism (uric acid), mineral metabolism (calcium, phosphorus, magnesium), and functional state of the liver (total bilirubin, ALT, AST, alkaline phosphatase) in the blood serum of animals was determined on an automatic biochemical analyzer “Konelab 20i” (ThermoScientific, Waltham, MA, USA). Fat was extracted from the liver using the Folch method [60]. The content of triglycerides and cholesterol in fat extracted from the liver was determined photometrically on an automatic biochemical analyzer Konelab 20i (ThermoScientific, Waltham, MA, USA).

#### 4.2.9. Blood Hematological Analysis

A general hematological analysis of samples of collected blood was carried out using a veterinary hemoanalyzer Exigo H400 (Boule, Spånga, Sweden). The following parameters were determined: the concentration of leukocytes, lymphocytes, monocytes, granulocytes, platelets, erythrocytes, the average erythrocyte volume, hemoglobin level, the average hemoglobin content in an erythrocyte, hematocrit, and width distribution of red blood cells by volume.

#### 4.2.10. Study of Microbiota

The microbiota was studied by real-time PCR using the Colonoflor 16 Premium test system (Alfa Labs, Saint-Petersburg, Russia).

DNA extraction from the contents of the cecum (0.1 g) was carried out using the DNA-Sorb-S kit (FBUN Central Research Institute of Epidemiology) with additional homogenization of samples in 2 mL microtubes containing lysis buffer from the extraction kit and 0.8 g sterile glass beads with a diameter of 0.1 mm. Amplification and detection were carried out using a CFX96 Real Time System amplifier (Bio-Rad, Hercules, CA, USA). Threshold cycle values (Cq) were calculated automatically by the CFX Manager software v. 3.1. The results were interpreted using the supplied Colonoflor software v. 2.1.5.0. The results were expressed as decimal logarithms of the number of genome-equivalent CFU/g. Additionally the ratio of log levels of *B. fragilis* group/*F. prausnitzii* was calculated.

### 4.3. Statistical Analysis

Statistical processing of the obtained results was carried out using the SPSS Statistics 20 (IBM) software. The sample was tested for normality using the Kolmogorov–Smirnov test at *p* = 0.05. In the case of normal distribution, parametric research methods were used: ANOVA according to plan 4^1^ (one-factor, four-level, balanced experiment). In case of rejection of the null hypothesis, the method of multiple comparison of means was used—Tukey’s test (q-test). The mean (M) and standard error of the mean (SEM) were calculated; data are presented as M ± SEM. If the sample did not correspond to a normal distribution, nonparametric goodness-of-fit tests were used: Kruskal–Wallis H test. If the null hypothesis was rejected, the multiple comparison method, Duncan’s test (MRT), was used. The median (Me), lower (Q1), and upper (Q3) quartiles were determined. Data are presented as Me (Q1–Q3). Differences were considered statistically significant at *p <* 0.05.

## 5. Conclusions

A feature of this study was the use of a hypercholesterolemia model in a subchronic experiment in comparison and in combination with forced running exercise. We found that the introduction of 2% cholesterol into the diet of animals has the greatest effect on the body of rats, without causing serious pathology or mortality, which is also confirmed by the successful correction of some indicators by running exercise. The data obtained may be of interest when modeling hyperlipidemia in rodents for the assessment of the hypolipidemic and hypocholesterolemic properties of both drugs and functional foods. This model can be used to reproduce non-alcoholic fatty liver disease in vivo firstly for testing target substances to correct changes occurring in the liver. The dysbiotic changes shown in the experiment, characteristic of a diet high in fats and simple sugars, can also be considered as a valid biomarker for possible dietary correction using functional foods. The combination of factors used to simulate poor nutrition against the background of regular moderate physical activity showed, first of all, the prospects for using physical activity in the prevention and treatment of the disease.

## Figures and Tables

**Figure 1 ijms-25-05383-f001:**
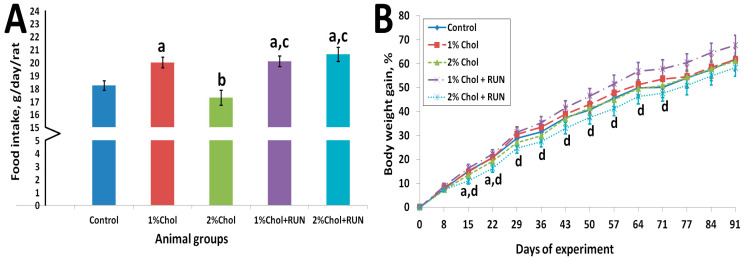
(**A**) Average food consumption, g/day/rat; (**B**) body weight gain, %. Note: a—differences are significant against Control group; b—differences are significant against 1% Chol group; c—differences are significant against 2% Chol group; d—differences are significant against 1% Chol + RUN group; *p <* 0.05 (Tukey’s test); error bars show the standard error of the mean.

**Figure 2 ijms-25-05383-f002:**
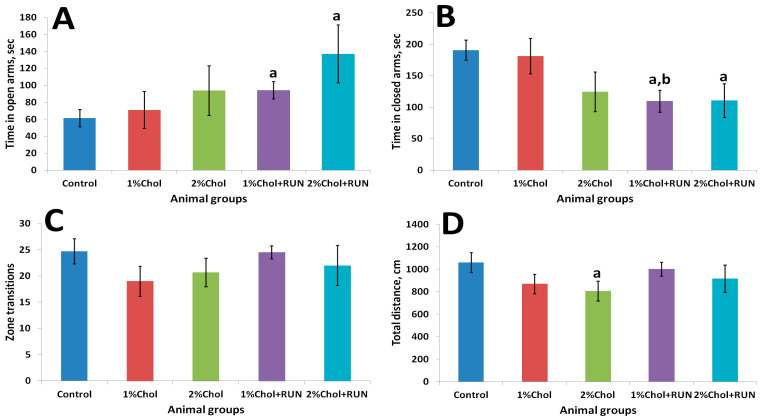
Elevated plus maze test results. (**A**)—time spent in open arms, s; (**B**)—time spent in closed arms, s; (**C**)—number of zone transitions; (**D**)—total distance travelled, cm. Note: a—differences are significant against Control group (*p <* 0.05); b—differences are significant against group 1% Chol (*p <* 0.05) Tukey’s test; error bars show the standard error of the mean.

**Figure 3 ijms-25-05383-f003:**
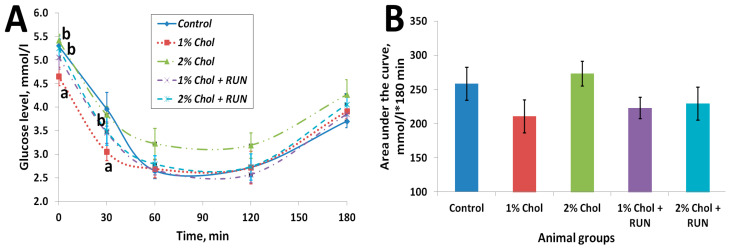
Results of insulin resistance test (**A**) dependence of glucose levels on time after insulin administration; (**B**) area under the curve, mmol/L × 180 min. Note: a—differences are significant against Control group (*p <* 0.05); b—differences are significant against group 1% Chol (*p <* 0.05); Tukey’s test; error bars show the standard error of the mean.

**Figure 4 ijms-25-05383-f004:**
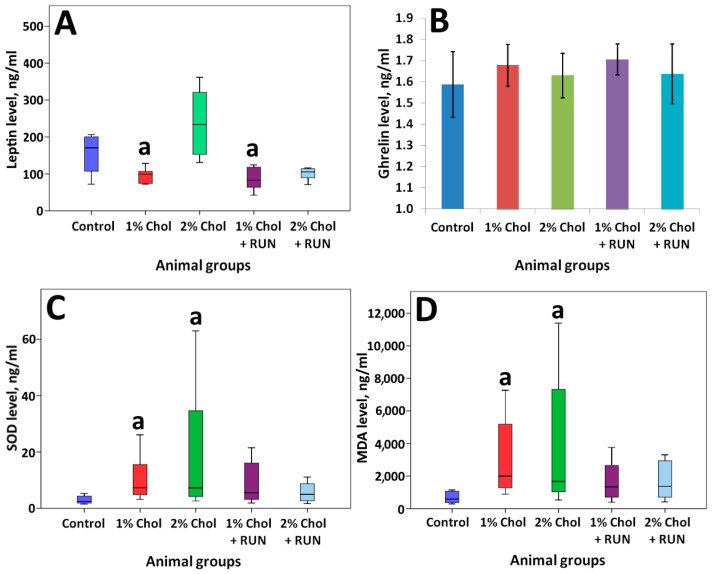
Serum biochemical parameters determined by ELISA. Note: leptin (**A**) MRT test, ghrelin (**B**) Tukey’s test, error bars show the standard error of the mean, SOD (**C**), MRT test; MDA (**D**) MRT test. a—differences are significant against Control group (*p <* 0.05). (**A**,**C**,**D**)—the bars show minimum and maximum values.

**Figure 5 ijms-25-05383-f005:**
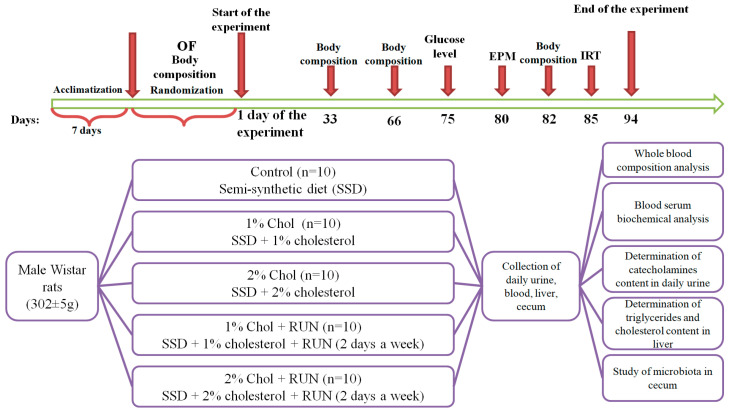
The experimental design. Note: OF—open field; EPM—elevated plus maze; IRT—insulin resistance test.

**Table 1 ijms-25-05383-t001:** Body composition of experimental animals.

Parameter	Animal Group				
	Control	1% Chol	2% Chol	1% Chol + RUN	2% Chol + RUN
Test 1 (before the start of the experiment)					
Fat content, %	9.2 ± 0.5	8.3 ± 0.5	10.0 ± 1.2	8.2 ± 0.7	9.8 ± 0.7
Lean weight, %	85.5 ± 0.5	86.4 ± 0.9	84.8 ± 1.2	86.8 ± 0.7	84.7 ± 0.6
Free water, %	0.25 ± 0.03	0.26 ± 0.02	0.32 ± 0.05	0.22 ± 0.01	0.22 ± 0.02
Total water, %	74.1 ± 0.5	74.7 ± 0.8	73.4 ± 1.0	74.9 ± 0.7	73.9 ± 0.6
Test 2 (33rd day)					
Fat content, %	9.5 ± 1.1	5.3 ± 0.8 ^a,^*	10.4 ± 2.02	5.1 ± 1.0 ^a,c,^*	6,5 ± 1,3 *
Lean weight, %	85.6 ± 1.1	90.2 ± 0.8 ^a,^*	85.0 ± 2.1 ^b^	90.6 ± 1.0 ^a,c,^*	88.6 ± 1.2 *
Free water, %	0.24 ± 0.02	0.27 ± 0.03	0.30 ± 0.03 ^a^	0.29 ± 0.02 ^a^	0.30 ± 0.02 ^a^
Total water, %	72.3 ± 1.0	76.1 ± 0.6 ^a^	70.7 ± 1.7 ^b^	76.5 ± 0.9 ^a,c^	75.0 ± 1.1
Test 3 (66th day)					
Fat content, %	10.3 ± 1.2	5.4 ± 0.7 ^a,^*	12.7 ± 2.6 ^b^	5.5 ± 1.2 ^a,c,^*	6.3 ± 1.4 ^a,c,^*
Lean weight, %	83.5 ± 1.2	88.9 ± 0.7 ^a,^*	81.3 ± 2.7 ^b^	89.1 ± 1.1 ^a,c^	88.0 ± 1.4 ^a,c,^*
Free water, %	0.18 ± 0.01	0.26 ± 0.05	0.20 ± 0.02	0.24 ± 0.05	0.19 ± 0.03
Total water, %	70.2 ± 0.9 *	74.7 ± 0.6 ^a^	68.4 ± 2.2 ^b^	75.1 ± 1.0 ^a,c^	74.1 ± 1.2 ^a,c^
Test 4 (82nd day)					
Fat content, %	10.5 ± 1.0	5.4 ± 0.7 ^a,c,^*	15.4 ± 2.2 ^a^	5.8 ± 1.3 ^a,c^	6.7 ± 1.5 ^a,c,^*
Lean weight, %	83.2 ± 1.0 *	88.6 ± 0.7 ^a,c^	78.2 ± 2.3 ^a^	88.2 ± 1.2 ^a,c^	87.1 ± 1.6 ^a,c^
Free water, %	0.27 ± 0.02	0.31 ± 0.03	0.27 ± 0.03	0.24 ± 0.02	0.24 ± 0.03
Total water, %	69.2 ± 1.0 *	74.5 ± 0.6 ^a,c^	67.1 ± 2.2 ^a,^*	74.5 ± 1.1 ^a,c^	73.3 ± 1.3 ^a,c^

Note: ^a^—differences are significant against Control group; ^b^—differences are significant against group 1% Chol; ^c^—differences are significant against group 2% Chol; *—differences are significant against test 1; *p <* 0.05.

**Table 2 ijms-25-05383-t002:** General biochemical analysis of blood serum.

Parameter	Animal Groups				
	Control	1% Chol	2% Chol	1% Chol + RUN	2% Chol + RUN
Total protein, g/L	75.0 [69.7–77.8]	77.3 [74.2–82.6]	82.1 [78.2–88.7] ^a^	76.4 [73.8–82.1]	81.9 [77.4–83.9] ^a^
Albumin, g/L	35.8 [35.5–36.0]	35.5 [35.1–36.8]	37.2 [35.9–38.1]	36.1 [35.6–36.9]	36.7 [36.0–37.4]
Globulins, g/L	38.2 ± 1.5	42.6 ± 1.6	45.7 ± 2.5 ^a^	41.7 ± 1.8	44.6 ± 1.5 ^a^
Cholesterol, mmol/L	1.5 [1.4–1.6]	1.8 [1.3–1.9]	2.8 [2.2–2.9] ^a,b^	1.8 [1.7–1.9] ^c^	2.2 [1.8–2.8] ^a,b^
Triglycerides, mmol/L	1.0 [0.8–2.3]	0.8 [0.7–1.1]	0.9 [0.7–1.6]	0.8 [0.6–0.9]	1.0 [0.6–1.1]
HDL, mmol/L	1.2 [1.1–1.4]	0.9 [0.8–1.0] ^a^	1.1 [1.1–1.2] ^b^	0.9 [0.8–0.9] ^a,c^	1.1 [1.0–1.2] ^b,d^
LDL, mmol/L	0.07 [0.05–0.08]	0.13 [0.09–0.27] ^a^	0.20 [0.11–0.24] ^a^	0.12 [0.10–0.14] ^a^	0.12 [0.08–0.21] ^a^
Atherogenic coefficient	0.23 [0.14–0.27]	0.85 [0.49–1.15] ^a^	1.20 [0.99–1.63] ^a^	1.03 [0.89–1.13] ^a^	1.07 [0.45–1.88] ^a^
Total bilirubin, μmol/L	4.27 ± 0.22	3.77 ± 0.14 ^a^	3.48 ± 0.11 ^a^	3.57 ± 0.07 ^a^	3.24 ± 0.22 ^a,b^
Creatinine, μmol/L	53.1 [51.3–56.1]	52.4 [51.7–53.1]	57.5 [53.1–59.0] ^b^	53.3 [53.0–56.1]	57.0 [55.3–58.6] ^a,b^
Alkaline phosphatase, U/L	101 [88–121]	115 [98–124]	123 [112–155]	117 [97–147]	162 [129–183] ^a^
ALT, U/L	56.6 [55.4–65.4]	56.4 [49.0–64.3]	80.7 [75.3–109.5] ^a,b^	59.7 [50.2–65.5] ^c^	91.7 [72.2–157.0] ^a,b,d^
AST, U/L	39.4 ± 3.9	53.5 ± 5.1	60.1 ± 4.6 ^a^	60.0 ± 7.2 ^a^	42.3 ± 4.9 ^c^
AST/ALT	0.70 ± 0.10	0.92 ± 0.14	0.67 ± 0.10	0.98 ± 0.09	0.45 ± 0.11 ^b,d^
Urea, mmol/L	5.0 ± 0.2	4.7 ± 0.1	5.1 ± 0.3	4.8 ± 0.2	5.0 ± 0.1
Uric acid, μmol/L	86.6 ± 3.4	91.1 ± 4.6	88.2 ± 5.5	83.1 ± 2.6	87.1 ± 3.4
Phosphor, mmol/L	2.27 ± 0.04	2.37 ± 0.06	2.28 ± 0.05	2.39 ± 0.09	2.45 ± 0.07
Magnesium, mmol/L	0.89 [0.87–0.96]	0.83 [0.76–0.86]	0.91 [0.86–0.96]	0.91 [0.83–0.99]	0.90 [0.90–0.92] ^b^
Calcium, mmol/L	2.95 [2.94–3.00]	3.01 [2.97–3.03]	3.05 [2.99–3.08] ^a^	3.02 [2.97–3.08]	3.16 [3.06–3.21] ^a,b,d^
Glucose, mmol/L	5.7 ± 0.3	5.7 ± 0.1	6.2 ± 0.3	5.7 ± 0.2	6.3 ± 0.2

Note: ^a^—differences are significant against Control group; ^b^—differences are significant against group 1% Chol; ^c^—differences are significant against group 2% Chol; ^d^—differences are significant against group 1% Chol + RUN; *p <* 0.05.

**Table 3 ijms-25-05383-t003:** Liver biochemical parameters (wet weight).

Parameter	Animal Groups				
	Control	1% Chol	2% Chol	1% Chol + RUN	2% Chol + RUN
Liver weight, g	11.4 [10.0–11.7]	14.3 [13.9–17.2] ^a^	14.2 [13.8–17.7] ^a^	15.1 [13.8–16.6] ^a^	14.5 [14.0–18.4] ^a^
Relative liver weight, %	2.3 [2.2–2.4]	3.0 [2.6–3.2] ^a^	3.2 [2.9–3.4] ^a^	2.9 [2.9–3.1] ^a^	3.0 [2.9–3.7] ^a^
Fat, mg/g of liver	66.4 [63.6–70.2]	138.7 [118.2–163.5] ^a^	154.5 [128.3–223.2] ^a^	150.4 [129.0–159.5] ^a^	158.2 [108.9–210.4] ^a^
Triglycerides, mg/g	6.6 [4.4–13.4]	31.2 [16.3–37.3] ^a^	31.4 [23.2–36.2] ^a^	18.8 [6.6–30.9]	20.4 [9.8–30.2] ^a^
Cholesterol, mg/g	3.0 [2.5–3.4]	20.6 [16.4–22.4] ^a^	23.6 [17.5–28.2] ^a^	18.6 [15.6–22.8] ^a^	17.5 [11.9–22.1] ^a^

Note: ^a^—differences are significant against Control group; *p <* 0.01.

**Table 4 ijms-25-05383-t004:** General hematological whole blood analysis.

Parameter	Animal Groups				
	Control	1% Chol	2% Chol	1% Chol + RUN	2% Chol + RUN
Leukocytes, 10^9^/L	5.5 [4.6–6.4]	6.5 [5.4–7.4]	4.3 [3.8–5.5] ^b^	7.8 [5.2–9.2] ^c^	5.9 [4.9–6.7] ^c^
Lymphocytes, 10^9^/L	3.6 [3.1–4.5]	4.0 [3.3–4.9]	2.8 [1.9–3.5] ^b^	4.8 [3.3–6.6] ^c^	4.0 [2.9–4.4]
Lymphocytes, %	66.2 ± 2.1	64.1 ± 2.8	61.5 ± 3.3	66.1 ± 2.3	61.6 ± 2.1
Monocytes, 10^9^/L	0.24 ± 0.03	0.36 ± 0.05	0.20 ± 0.05 ^b^	0.26 ± 0.03	0.31 ± 0.05
Monocytes, %	4.3 [3.8–7.3]	7.2 [5.1–7.7]	4.5 [4.0–7.6]	3.8 [3.3–4.9]	7.3 [3.7–8.1]
Granulocytes, 10^9^/L	1.6 [1.2–1.8]	2.1 [1.3–2.3]	1.1 [0.9–1.5] ^b^	2.0 [1.7–2.3] ^c^	1.7 [1.5–2.0]
Granulocytes, %	29.7 [23.6–34.7]	28.9 [27.3–34.5]	28.4 [22.4–43.4]	29.9 [24.5–33.8]	30.6 [28.5–35.6]
Hemoglobin, g/L	153 [151–157]	146 [146–150]	150 [147–152]	150 [144–155]	155 [154–156] b
Average hemoglobin content in an erythrocyte, pg	18.7 ± 0.2	18.4 ± 0.3	17.9 ± 0.2 ^a^	18.6 ± 0.1 ^c^	18.7 ± 0.2 ^c^
Average concentration of cellular hemoglobin in erythrocytes, g/L	365.7 ± 2.4	374.8 ± 1.6	371.2 ± 1.5	370.1 ± 1.4	367.4 ± 1.0
Erythrocytes, 10^12^/L	8.3 [8.0–8.5]	8.1 [7.9–8.2]	8.4 [8.2–8.5]	7.9 [7.8–8.3]	8.3 [8.2–8.3]
Average erythrocyte volume, fl	50.6 [50.0–51.9]	48.9 [47.4–50.9]	48.0 [47.4–48.7] ^a,b^	50.6 [49.6–51.1] ^c^	51.2 [50.0–51.7] ^b,c^
Hematocrit, %	42.2 ± 0.6	39.7 ± 0.7 ^a^	40.3 ± 0.3 ^a^	40.3 ± 0.6 ^a^	42.1 ± 0.5 ^b,c,d^
Width of distribution of red blood cells by volume, %	13.0 [12.8–13.5]	13.2 [13.1–13.7]	13.7 [13.2–14.0]	13.3 [13.1–13.5]	13.1 [12.7–13.6]
Width of distribution of red blood cells by volume, fl	31.2 ± 0.4	29.9 ± 0.4	29.9 ± 0.4	30.7 ± 0.2	31.3 ± 0.4
Platelets, 10^9^/L	705.3 ± 71.2	738.7 ± 73.8	726.0 ± 63.5	761.6 ± 54.9	705.6 ± 45.7
Platelets average volume, fl	6.39 ± 0.14	6.20 ± 0.10	6.10 ± 0.10	6.21 ± 0.09	6.33 ± 0.07

Note: fl—femtoliter (1 × 10^−15^); ^a^—differences are significant against Control group; ^b^—differences are significant against group 1% Chol; ^c^—differences are significant against group 2% Chol; ^d^—differences are significant against group 1% Chol + RUN; *p <* 0.05.

**Table 5 ijms-25-05383-t005:** Levels of microorganisms in the contents of the cecum (lg genome-eq/g).

Populations and Groups of Microorganisms	Control		1% Chol		2% Chol		1% Chol + RUN		2% Chol + RUN	
	Me (Q1–Q3)	% Detected	Me (Q1–Q3)	% Detected	Me (Q1–Q3)	% Detected	Me (Q1–Q3)	% Detected	Me (Q1–Q3)	% Detected
Total bacterial mass	12.9 [12.8–13.0]	100	13.0 [13.0–13.2]	100	12.8 [12.8–13.0]	83	12.9 [12.8–13.0]	100	12.9 [12.7–13.0]	100
*Lactobacillus* spp.	7.8 [7.6–8.5]	100	8.7 [8.4–9.3]	100	9.0 ^a^ [8.8–9.4]	100	9.2 ^a^ [8.8–9.4]	100	9.0 ^a^ [8.8–9.2]	100
*Bifidobacterium* spp.	10.5 [8.9–11.5]	100	8.7 [8.5–8.8]	100	8.5 [8.1–8.8]	100	8.3 ^a^ [8.3–8.7]	100	8.3 ^a^ [8.1–8.4]	100
*Escherichia coli*	6.8 [6.6–7.6]	100	8.5 ^a^ [8.2–8.9]	100	8.2 ^a^ [8.0–8.3]	100	8.5 ^a^ [7.3–8.7]	100	8.0 [7.9–8.2]	100
*Bacteroides* spp.	11.7 [11.4–11.9]	100	12.3 [12.0–12.3]	100	11.9 [11.6–11.9]	100	11.8 [11.7–12.0]	100	11.8 [11.6–12.0]	100
*Faecalibacterium prausnitzii*	9.0 [8.8–9.2]	100	8.0 ^a^ [8.0–8.2]	100	8.2 ^a^ [7.9–8.5]	100	8.7 ^b^ [8.4–8.8]	100	8.2 ^a^ [7.8–8.5]	100
*Enterococcus* spp.	5.7 [5.6–5.7]	33	7.3 [7.3–7.3]	17	–	0	7.2 [6.0–8.5]	33	–	0
*Staphylococcus aureus*	–	0	7.5 [6.7–8.2]	75	6.5 [6.5–6.5]	33	–	0	6.5 [6.5–6.5]	33
*Proteus vulgaris/mirabilis*	–	0	–	0	–	0	5.8 [5.8–5.8]	33	–	0
*Blautia* spp.	10.4 [10.3–10.8]	100	11.5 ^a^ [11.4–11.8]	100	11.5 [11.1–11.5]	100	11.3 [11.1–11.4]	100	11.1 [10.9–11.5]	100
*Acinetobacter* spp.	8.5 [8.3–8.6]	100	8.6 [8.5–8.8]	100	8.2 ^a,b^ [8.0–8.3]	100	8.3 ^b^ [8.1–8.3]	100	8.0 ^a,b^ [8.0–8.0]	100
*Streptococcus* spp.	7.5 [7.2–7.8]	67	7.7 [7.7–7.8]	100	7.6 [7.3–7.7]	100	7.3 [7.1–7.7]	100	7.4 ^b^ [7.3–7.6]	100
*Roseburia inulinivorans*	8.5 [8.5–8.7]	100	8.7 [8.6–8.8]	100	8.7 [8.5–8.8]	100	8.6 [8.5–8.8]	100	8.7 [8.6–8.9]	100
*Prevotella* spp.	8.0 [7.1–8.5]	67	7.5 [7.2–7.7]	50	6.8 [6.6–6.9]	100	6.3 [5.7–7.8]	83	6.7 [6.4–6.9]	100
*Ruminococcus* spp.	7.7 [7.7–7.7]	17	9.8 ^a^ [9.4–11.5]	83	8.7 [8.6–8.7]	33	10.5 [10.2–10.9]	33	8.5 [8.2–10.2]	50
*Bacteroides* spp. and *F. prausnitzii* (Bfr/Fprau) ratio	338 [133.3–1361.1]	–	12,571 ^a^ [4285–20,000]	–	5476 ^a^ [2821–8416]	–	2500 ^b^ [1160–3125]	–	4166 ^a^ [1833–8750]	–

Note: ^a^—differences are significant against Control group; ^b^—differences are significant against group 1% Chol.

**Table 6 ijms-25-05383-t006:** Division of animals into groups.

Parameter	Animal Group				
	Control	1% Chol	2% Chol	1% Chol + RUN	2% Chol + RUN
Body weight, g	302.2 ± 4.0	302.2 ± 6.4	302.0 ± 6.5	302.9 ± 5.7	302.1 ± 3.5
Time in center, s	5.9 ± 1.8	5.7 ± 1.8	6.1 ± 2.2	7.2 ± 2.2	5.6 ± 1.4
Zone transitions	10.7 ± 1.9	12.9 ± 1.6	12.6 ± 2.3	12.6 ± 2.0	11.6 ± 1.7
Distance, s	1764 ± 92	1762 ± 112	1783 ± 105	1720 ± 94	1692 ± 135

**Table 7 ijms-25-05383-t007:** Scheme of running training for animals of groups 1% Chol + RUN and 2% Chol + RUN.

Days of Experiment	Speed. cm/s	Time. min	Track Slope. Degrees
9 11 16 18	20	10	0
23 25 30 32	24	10	0
37 39 44 46 51 53	26	10	0
58 60 65 67 71 74 79	28	10	0
81 85 88 92 93	30	10	0

## Data Availability

The data presented in this study are available on request from the corresponding author.

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
