# Peer review of "The Influence of a High-Cholesterol Diet and Forced Training on Lipid Metabolism and Intestinal Microbiota in Male Wistar Rats"

_ijms, 2024, doi:10.3390/ijms25105383_

Round 1

Reviewer 1 Report

Comments and Suggestions for Authors

The paper: „The influence of a high-cholesterol diet and forced training on lipid metabolism and intestinal microbiota in male Wistar rats” gives a new approach of the influence of high-cholesterol diet and physical activity on both biochemical and blood parameters as well as gut microflora in animal model. IT is an important preclinical study which is original and novel.

The experiments are well-designed and there is a comprehensive approach in the possible changes in both glucose and lipid metabolism, blood tests, liver fat, weight and body composition as well as the intestinal flora due to high-cholesterol diet in animal model. The conclusions are valid and the discussion is well-written.

Some minor issues should be addressed before the paper is accepted for publication.

1.     The abstract appears unfinished since there are no data regarding the findings of the gut microflora changes due to the specific diet, although the intestinal microflora is mentioned in the title, nor there is a conclusion. Moreover, the explanation “39 indicators are analyzed” is very general and needs to be more specified – biochemical analysis, weight and body composition, gut microflora etc. The more informative is the abstract the more probable is that the attention of the readers will be captured.

2.     Introduction – lines 63-66 should be rewritten since there is missing point. If the authors aimed to present the limitations and the advantages on studying diet influence on hypercholesterolemia with and without physical activity in animal model, it should be presented more comprehensively with clear points.

3.     Line 299 – the sentence “The main limitation…” should start without that part “the main limitation” since that lack of “changes in body weight and body weight gain of animals” is not the main limitation of the study. Later, before the conclusion the authors may emphasize the main limitations of the study, including the general limitation to extrapolate these results from animal model to humans, as well as to point out the advantages of the study.

4.     There is lack of discussion in which model is explained as suitable for studying NAFLD in rodents although later in the conclusion it is said “This model can be effectively used to reproduce non-alcoholic fatty liver disease in vivo to find ways to correct it.” So please expand the discussion rather in this way after lines 346-354.

Author Response

The authors thank the Reviewer for an objective review of the article and for the opportunity to present it again, taking into account the comments made. The responses to the Reviewer's comments and recommendations are shown further. Major changes have been made to the text of the article and are highlighted in red.

  1. The abstract appears unfinished since there are no data regarding the findings of the gut microflora changes due to the specific diet, although the intestinal microflora is mentioned in the title, nor there is a conclusion. Moreover, the explanation “39 indicators are analyzed” is very general and needs to be more specified – biochemical analysis, weight and body composition, gut microflora etc. The more informative is the abstract the more probable is that the attention of the readers will be captured.

The abstract was rewritten according to Reviewer`s recommendations.

  1. Introduction – lines 63-66 should be rewritten since there is missing point. If the authors aimed to present the limitations and the advantages on studying diet influence on hypercholesterolemia with and without physical activity in animal model, it should be presented more comprehensively with clear points.

The sentence was corrected according to your recommendations.

  1. Line 299 – the sentence “The main limitation…” should start without that part “the main limitation” since that lack of “changes in body weight and body weight gain of animals” is not the main limitation of the study. Later, before the conclusion the authors may emphasize the main limitations of the study, including the general limitation to extrapolate these results from animal model to humans, as well as to point out the advantages of the study.

The sentence was corrected according to your recommendations.

  1. There is lack of discussion in which model is explained as suitable for studying NAFLD in rodents although later in the conclusion it is said “This model can be effectively used to reproduce non-alcoholic fatty liver disease in vivo to find ways to correct it.” So please expand the discussion rather in this way after lines 346-354.

The discussion was expanded.

Reviewer 2 Report

Comments and Suggestions for Authors

The article entitled “The influence of a high-cholesterol diet and forced training on lipid metabolism and intestinal microbiota in male Wistar rats” by Sidorova et al deals with the comparative assessments of cholesterol diet and physical activity on lipid metabolism /microbiota in male Wistar rats. Although the study is interesting from the point of lipid science there are several concerns authors need to address before considering it for publication.

Abstract: The background is missing also it seems that the abstract is mainly the methods part highlighted lacking the specific results of the study and the conclusion is also missing. I recommend authors rewrite their abstract. What is RUN does it mean running?

Introduction: It would be more fascinating if the author could consider giving background about the lipids changes and microbiota impacts of  in the introduction section.

Figures 1, 2 and 3 what is the error bar SE or SD could you mention it in legends? Also, what type of statistical test was applied should be mentioned in the legend.

Table 1: statistical test data is missing.

Line 164; it should be 82nd

Table 3: is it plasma data? Be more specific about the type of sample used for analysis.

Could authors explain compared to 1%Chol group why 1% Chol+RUN has higher liver weight?

The whole methodology is very much unclear and difficult to follow.  The recommended authors should provide detailed experimental information in the methods section for all the experiments they conducted, such that this study can be replicated. Although authors shows many results, their experimental details are missing.

Comments on the Quality of English Language

Nil

Author Response

The authors thank the Reviewer for an objective review of the article and for the opportunity to present it again, taking into account the comments made. The responses to the Reviewer's comments and recommendations are shown further. Major changes have been made to the text of the article and are highlighted in red.

Abstract: The background is missing also it seems that the abstract is mainly the methods part highlighted lacking the specific results of the study and the conclusion is also missing. I recommend authors rewrite their abstract. What is RUN does it mean running?

The abstract was rewritten according to Reviewer`s recommendations.

Introduction: It would be more fascinating if the author could consider giving background about the lipids changes and microbiota impacts of  in the introduction section.

The information is added to Introduction section.

Figures 1, 2 and 3 what is the error bar SE or SD could you mention it in legends? Also, what type of statistical test was applied should be mentioned in the legend.

The error bars present the standard error of the mean. This information was added to the notes of the figures. The applied statistical tests were also added to the notes of the figures.

Table 1: statistical test data is missing.

The notes for the table 1 were corrected, excess information was deleted.

Line 164; it should be 82nd

The number was corrected.

Table 3: is it plasma data? Be more specific about the type of sample used for analysis.

The data in the table is presented for blood serum. The heading of the table is rewritten. The information on sample type was added to the text, where it was missing.

Could authors explain compared to 1%Chol group why 1% Chol+RUN has higher liver weight?

The difference in absolute values of liver weight between these groups (not significant, p=0,66) is due to higher, but also not significant, body weight of the animals, exposed to running. Respectively the relative liver weight of 1% Chol and 1% Chol + RUN was 3,0% and 2.9% (p= 0,97), the differences are not significant according MRT test.

The whole methodology is very much unclear and difficult to follow.  The recommended authors should provide detailed experimental information in the methods section for all the experiments they conducted, such that this study can be replicated. Although authors shows many results, their experimental details are missing.

Authors checked the Material and methods and Results sections and structured them for easy understanding of the material.

Round 2

Reviewer 2 Report

Comments and Suggestions for Authors

The authors addressed the concerns sufficiently, however, there are still some corrections that may be required, as mentioned below.

-Improve the resolution of Fig 1. Why are axes labels in italics? Also, it is unclear about the number mentioned in Fig 1A (1,2, 1,3) and Fig 1B (1, 1,4, 4,4,4,).

-Remove grid lines, and also I recommend showing the significant differences in either with * or a, b,c, with proper description in the legend.

-Remove grid lines in Fig 3. nmol/l or nmol/L unify them. (the Y-axes and legend units are different)

-Table 75,0 should be 75.0 Replace “,” with “.” In the whole table. Align the tables properly.  The numbers are misplaced.

Fig 4B: remove grid lines and improve resolution and axes labels.

Comments on the Quality of English Language

Check the manuscript with a native English speaker is recommended.

Author Response

The authors addressed the concerns sufficiently, however, there are still some corrections that may be required, as mentioned below.

-Improve the resolution of Fig 1. Why are axes labels in italics? Also, it is unclear about the number mentioned in Fig 1A (1,2, 1,3) and Fig 1B (1, 1,4, 4,4,4,).

The resolution of all figures was improved. The italics was removed. The numbers for differences were changed to letters (a, b, c, d). We made changes to the legend, which are marked in red.

-Remove grid lines, and also I recommend showing the significant differences in either with * or a, b,c, with proper description in the legend.

The grid lines were removed. The significant differences were changed to letters (a, b,c, d).

-Remove grid lines in Fig 3. nmol/l or nmol/L unify them. (the Y-axes and legend units are different).

The grid lines were removed. The typo was corrected to nmol/l in legend.

-Table 75,0 should be 75.0 Replace “,” with “.” In the whole table. Align the tables properly.  The numbers are misplaced.

All tables were corrected.

Fig 4B: remove grid lines and improve resolution and axes labels.

The resolution of all figures was improved. The grid lines were removed.